# Development of a Benzophenone-Free Red Propolis Extract and Evaluation of Its Efficacy against Colon Carcinogenesis

**DOI:** 10.3390/ph17101340

**Published:** 2024-10-08

**Authors:** Iara Silva Squarisi, Victor Pena Ribeiro, Arthur Barcelos Ribeiro, Letícia Teixeira Marcos de Souza, Marcela de Melo Junqueira, Kátia Mara de Oliveira, Gaelle Hayot, Thomas Dickmeis, Jairo Kenupp Bastos, Rodrigo Cassio Sola Veneziani, Sérgio Ricardo Ambrósio, Denise Crispim Tavares

**Affiliations:** 1Research Group on Natural Products, Center for Research in Sciences and Technology, University of Franca, Franca 14404-600, SP, Brazil; iasquarisi79@outlook.com (I.S.S.); vpribeir@olemiss.edu (V.P.R.); arthur1_@hotmail.com (A.B.R.); leticiateixeiramds@gmail.com (L.T.M.d.S.); marcela.junqueira@cs.unifran.edu.br (M.d.M.J.); kmoliveiraq@gmail.com (K.M.d.O.); rodrigo.veneziani@unifran.edu.br (R.C.S.V.); 2Institute of Biological and Chemical Systems—Biological Information Processing—Karlsruhe Institute of Technology, 76131 Karlsruhe, Germany; gaelle.hayot@kit.edu (G.H.);; 3School of Pharmaceutical Sciences of Ribeirão Preto, University of São Paulo, Ribeirão Preto 14040-903, SP, Brazil; jkbastos@fcfrp.usp.br

**Keywords:** red propolis, benzophenone free, zebrafish, genotoxicity, colon cancer

## Abstract

Background/Objectives: Brazilian red propolis has attracted attention for its pharmacological properties. However, signs of toxicity were recently observed in long-term studies using the hydroalcoholic extract of red propolis (RPHE), likely due to polyprenylated benzophenones. This study aimed to develop a benzophenone-free red propolis extract (BFRP) and validate an HPLC-PDA method to quantify its main constituents: isoliquiritigenin, vestitol, neovestitol, medicarpine, and 7-*O*-methylvestitol. Methods: BFRP’s toxicity was assessed in zebrafish larvae through a vibrational startle response assay (VSRA) and morphological analysis. Genotoxicity was evaluated using the micronucleus test in rodents, and the extract’s effects on chemically induced preneoplastic lesions in rat colon were studied. An HPLC-PDA method was used to quantify BFRP’s main compounds. Results: BFRP primarily contained vestitol (128.24 ± 1.01 μg/mL) along with isoliquiritigenin, medicarpin, neovestitol, and 7-O-methylvestitol. Zebrafish larvae exposed to 40 µg/mL of BFRP exhibited toxicity, higher than the 10 µg/mL for RPHE, though no morphological differences were found. Fluorescent staining in the notochord, branchial arches, and mouth was observed in larvae treated with both BFRP and RPHE. No genotoxic or cytotoxic effects were observed up to 2000 mg/kg in rodents, with no impact on hepatotoxicity or nephrotoxicity markers. Chemoprevention studies showed a 41.6% reduction in preneoplastic lesions in rats treated with 6 mg/kg of BFRP. Conclusions: These findings indicate that BFRP is a safe, effective propolis-based extract with potential applications for human health, demonstrating reduced toxicity and chemopreventive properties.

## 1. Introduction

Propolis is a natural product of beekeeping widely used by humanity as a natural medicine, a supplement in the food industry, and an adjuvant in the cosmetic industry. Among the thirteen types of Brazilian propolis classified according to their physiochemical characteristics, the red propolis produced in the mangroves of northeast Brazil was the last identified [1] and is regarded as the second most significant type of Brazilian propolis, with increasing exports to various countries [2].

Since 2007, when red propolis was identified, studies have shown a variety of biological activities associated with this bee product [3], such as antioxidant [4], antibacterial [5,6,7], anti-periodontopathogenic bacteria, and anti-*Chikungunya* virus [8] activities; cutaneous wound healing [9]; and immunomodulatory, antiproliferative [10,11,12], anticarcinogenic [13], and gastroprotective [14,15] properties.

Chemically, red propolis is a rich source of phenolic compounds such as chalcones, flavonoids, isoflavones, isoflavans, and pterocarpans. These phenolic compounds originate from its botanical source *Dalbergia ecastaphyllum*, which is responsible for providing compounds such as liquiritigenin, isoliquiritigenin, formononetin, vestitol, neovestitol, medicarpine, and methylvestitol [16,17]. Moreover, phytochemical studies have also reported the presence of polyprenylated benzophenones, such as guttiferone E, xanthochymol, and oblongifolin B, whose source is *Symphonia globulifera* [18].

Although red propolis demonstrates promising biological activities, its potential advantages over conventional cancer therapies warrant further investigation. Chemotherapeutic agents are vital for cancer treatment but often come with significant side effects due to their non-specific targeting. These drugs can induce systemic toxicity by affecting normal cells, leading to problems such as anemia, hair loss, and gastrointestinal issues [19]. This highlights the urgent need for alternative therapeutic strategies. 

Chemoprevention, which involves using agents like natural products to prevent cancer, plays a crucial role in reducing disease incidence. By focusing on prevention, these agents can minimize the necessity for aggressive treatments and lessen the overall cancer burden on patients [20]. Red propolis, with its bioactive compounds could offers a promising and less toxic alternative to conventional therapies, which are often associated with severe side effects.

Our research aims to develop a benzophenones-free red propolis extract (BFRP) to address the toxicity concerns related to the hydroalcoholic extract (RPHE) [21]. This new extract is intended to explore its potential in preventing colon carcinogenesis. By enhancing the safety and efficacy profile of red propolis, BFRP could become a valuable option for integrative oncology, appealing to patients who seek treatments with fewer side effects.

## 2. Results and Discussion

### 2.1. Obtaining the Standard Compounds and the Development and Validation of the RP-HPLC-PDA Method for the BFRP Analysis

As previously described, the compounds isoliquiritigenin, vestitol, neovestitol, medi-carpine, and 7-*O*-methylvestitol were isolated from the BFRP, and their chemical structures were confirmed by the NMR data in comparison with the literature [18,22] (Figure 1). It is important to mention that during the isolation and analysis processes, the presence of polyprenylated benzophenones was not identified, thus denoting that the goal of obtaining a benzophenone-free red propolis extract was achieved. This fact can also be confirmed by analyzing the chemical profile displayed by the BFRP chromatogram obtained through the analysis performed according to the analytical method whose development is described herein (Figure 2). An analysis of the hydroalcoholic extract and the benzophenones-free red propolis extract was presented in Appendix A.

Propolis is a complex natural product, abundant in flavonoids and other phenolic compounds. For phenolic compounds with strong chromophore groups, UV-based detectors provide excellent selectivity and sensitivity for the identification and quantification of bioactive compounds, making them an ideal choice for developing quality control methods. During the analytical method development, parameters such as temperature, flow rate, gradient, chromatographic conditions, including column and mobile phase, were optimized to achieve effective peak separation. The best peak resolution was obtained with a 30-min analysis (Figure 2). All parameters established during the method validation process comply with the relevant validation guidelines [23].

The UV spectra of each peak corresponded perfectly with their respective standards, confirming that each analytical signal aligns with its chromatographic counterpart. To assess linearity, standard solutions at eight different concentrations were analyzed using the HPLC-PDA method, yielding correlation coefficients (R) and determination coefficients (R²) derived from the analytical curves, which were statistically analyzed using the least squares method. The R² values for all analytical curves exceeded 0.99, demonstrating linearity in accordance with validation guidelines. Furthermore, residual analyses were conducted, confirming the homoscedasticity of the data, as the analytical curves exhibited no lack of fit (*p* > 0.05). The intervals obtained for the limits of detection and quantification were also satisfactory, indicating that the chromatographic method can effectively detect and quantify the selected compounds at low concentrations (Table 1).

The accuracy of the method was evaluated for both repeatability (intraday accuracy) and intermediate accuracy (interday accuracy). Results demonstrated minimal variation between analyses, with both intraday and interday precisions exhibiting relative standard deviations (RSDs) below 2.99%. Accuracy results were approximately 100%, ranging from 96.05% to 113.93%, indicating that the analytical method yielded precise results with errors below 4.07% (Table 2). Consequently, the developed method is considered accurate and precise, adhering to ANVISA guidelines. For the recovery assessment, a Soxhlet appa-ratus was utilized for the exhaustive extraction of red propolis, yielding an extract yield of 52.6%. Recovery rates for all evaluated analytes were found to be between 96% and 108% at low, medium, and high concentrations. Thus, the optimized method can be deemed reliable for extracting chemically similar phenolic compounds. Additionally, the recovery of the internal standard was also high, ranging from 96% to 100%.

The robustness was evaluated by the small variations in some parameters of the two-level method. The changes in these parameters led to a variation between 7 and 9% in the concentration of the compounds analyzed by the HPLC-PDA method. Staying within the limits established by the guides, therefore, the developed method proved to be robust.

Finally, it is possible to affirm that the developed method was successfully validated according to the guidelines of the Brazilian regulatory agency (ANVISA, Brasília, Brazil). Furthermore, the analyses performed using this method found that the phenolic compound content in the BFRP samples showed a predominant occurrence of vestitol (128.24 ± 1.01 μg/mL) in the extract, followed by isoliquiritigenin (87.94 ± 0.12 μg/mL), medicarpin (61.53 ± 0.42 μg/mL), neovestitol (53.36 ± 0.28 μg/mL), and 7-*O*-methylvestitol (29.69 ± 0.96 μg/mL). It is interesting to notice that vestitol was also identified as the major compound in the RPHE [17], although at a much lower concentration (65.06 ± 0.1796 μg/mL) than our findings in the BFRP. In fact, the concentration of the other main constituents of the BFRP that were previously quantified in the RPHE were also found at lower concentrations in the ethanolic extract (medicarpin: 48.72 ± 0.18 μg/mL; neovestitol: 33.03 ± 0.70 μg/mL; and 7-*O*-methylvestitol: 16.12 ± 0.29 μg/mL). These variations in the contents may be attributed to differences in the selectivity of the extraction process employed; however, other factors such as collection location and time of year, among others, could also be an influence.

### 2.2. Toxicity Assessment

#### 2.2.1. Fish Toxicity Test

The toxicities of the BFRP and RPHE were initially evaluated in the zebrafish larvae using the VSRA assay. In this species, the startle response to a vibration stimulus starts at 5 dpf [24]. The larvae were exposed to the BFRP and RPHE at concentrations ranging from 2.5 to 160 µg/mL from 3 dpf to 5 dpf (48 h treatment), and their responses to a vibration stimulus are shown in Figure 3. Concentrations greater than or equal to 40 µg/mL of the BFRP (Figure 3A) and 10 µg/mL of the RPHE (Figure 3B) caused immobility in all the larvae, indicating toxicity at these concentrations (*p* < 0.0001). The larvae treated with BFRP and RPHE concentrations lower than or equal to 20 and 5 µg/mL, respectively, did not show significant immobilization when compared to the untreated larvae.

The lower toxicity of the BFRP when compared to the RPHE is related to the absence of polyprenylated benzophenones. In view of the known toxicity of these compounds [17,25,26], the BFRP was developed to remove the polyprenylated benzophenones with the aim of reducing the toxicity of the red propolis.

Previously, our research group reported the toxicity of RPHE to adult zebrafish at concentrations greater than or equal to 12.5 µg/mL after 96 h of exposure [17]. The data obtained in this present study revealed the greater sensitivity of zebrafish larvae to the effects of RPHE compared to adults, since its toxicity was observed in larvae at a lower concentration (10 µg/mL) and with a shorter exposure time (24 h).

The zebrafish larvae treated with 40 µg/mL of the BFRP, as well as those treated with the same concentration of the RPHE, exhibited an altered swimming behavior after 4 h of treatment. Specifically, the larvae were swimming upside down at the bottom of the plate. This behavior indicates that the central nervous system, the inner ear, or the swim bladder may have been altered by exposure to the extracts, compromising swimming ability and leading to the death of the individual.

In order to better understand the altered swimming behavior, morphological analyzes on these zebrafish larvae were conducted. The larvae treated with the BFRP and RPHE did not exhibit morphological abnormalities regarding the evaluated parameters when compared to those not treated (Figure 4B–H). Therefore, the altered swimming behavior induced by the extracts was not related to morphological changes in the inner ear or the swim bladder. It is noteworthy that the function of these organs may have been altered by the exposure of the larvae to the BFRP and RPHE, although no morphological changes were observed. Further studies should be carried out to clarify the abnormal swimming behavior of the larvae induced by the extracts.

Furthermore, a fluorescence microscopy analysis of these larvae was carried out. Interestingly, for the first time, it is being reported that the larvae treated with the BFRP or RPHE exhibited fluorescent staining in the notochord, branchial arches, and mouth, compared to the untreated larvae (Figure 5). The fluorescence displayed in these organs after the treatment with the extracts, especially of the notochord, may be involved in the altered swimming behavior.

Additional studies are necessary to understand the relationship between the structures that exhibited fluorescence, and the toxicity observed in the larvae treated with the BFRP and RHPE. However, the mineralization of the notochord begins in the larval stage used in the experiments at 5 dpf [28]. Therefore, it can be speculated that the chemical constituents of the BFRP and RHPE have an affinity for calcium.

A segmented vertebral column is a distinctive feature of vertebrate species. In all vertebrates, somites are formed rhythmically and sequentially along the embryonic axis in a process regulated by a multicellular genetic oscillator known as the segmentation clock. This segmentation clock comprises a network of oscillating genes that exhibit variations among species [29]. It has been proposed that, in zebrafish [30] and salmon [31,32], the notochord rather than the sclerotome is the initial source of the bone matrix for cordacentra (vertebral body) formation, while in other species of teleost fish, the classic osteoblasts derived from the sclerotome have been suggested as the main drivers of the cordacentra formation [33,34]. The sclerotome occupies most of the somite, while skin and muscle are derived from the smaller dermatome. The cells within the sclerotome responsible for the production and mineralization of the organic aspect of bone (osteoid) are called osteoblasts. Cordacentra are first observed as they mineralize into rings around the notochord, forming sequentially along the axis in a process that begins several days after somitogenesis and muscle segmentation [28]. Thus, the beginning of the segmentation of the cordacentra can be observed through the fluorescence displayed by the BFRP and RPHE.

Among the flavonoids present in the BFRP and RPHE, isoliquiritigenin (ISL) is the only one reported in the literature to fluoresce [35]. The toxic effects of ISL have been reported on the development of zebrafish embryos (24 hpf) and larvae (96 hpf), especially on the heart, liver, and nervous system [36]. Nevertheless, the BFRP and RPHE are mixtures of substances. Therefore, studies with their substances isolated must be carried out, as well as molecular genetic analyses, to better understand the effects of the BFRP and RHPE on swimming behavior and the fluorescence displayed, especially by the notochord of zebrafish larvae.

#### 2.2.2. Genetic Toxicology Test

The results obtained by the micronucleus assay in the Swiss mice’s peripheral blood demonstrated that the frequencies of the MNPCEs of the animals treated with different doses of the BFRP (500, 1000, and 2000 mg/kg) did not differ significantly from the solvent group. In addition, the PCE/PCE + NCE (normochromatic erythrocyte) ratios were not significantly different between the treatment groups (Table 3). Therefore, the data revealed that the BFRP did not present genotoxicity and cytotoxicity, following the experimental recommendations of the OECD [37].

The animals submitted to the different treatments presented serum levels of ALT (Figure 6A) and creatinine (Figure 6B) that did not differ from those observed in the negative control group, revealing the absence of hepatotoxicity and nephrotoxicity, respectively.

Genotoxicity tests are designed to identify the compounds capable of causing genetic damage through various mechanisms. These assessments facilitate the recognition of potential dangers related to DNA damage and its fixation. The fixation of DNA damage, such as genetic mutations; extensive chromosomal damage; or recombination is generally considered crucial in the intricate process of malignancy, where genetic alterations can play a contributory role. Numerical changes in the chromosomes have also been associated with tumorigenesis and may indicate susceptibility to aneuploidy in germ cells. The compounds that produce positive results in tests that detect these types of damage have the potential to act as human carcinogens and/or mutagens. Therefore, genotoxicity tests mainly serve to predict carcinogenicity [38].

The micronucleus test used in this present study is part of the standard genetic toxicology battery for the prediction of potential risks for pharmaceuticals intended for human use. Negative results, as obtained for the BFRP, are considered sufficient to demonstrate the absence of a significant genotoxic risk [39].

It is important to consider that, although BFRP did not show genotoxicity in acute exposure, the long-term use of propolis, particularly over extended periods, may still pose potential health risks [40]. Prolonged exposure to the bioactive compounds in propolis, whether through internal or external application, could lead to cumulative effects that might not be immediately apparent in short-term genotoxicity assays. For instance, the risk of allergic reactions, sensitization, and potential interactions with other med-ications may increase over time, especially in susceptible individuals [41] 

Therefore, despite the absence of significant genotoxicity observed for BFRP, caution should be exercised with the long-term use of propolis, and further studies are recommended to thoroughly assess its safety profile under prolonged exposure.

### 2.3. Evaluation of the Effect of BFRP on Preneoplastic Colon Lesions

Preneoplastic lesions were observed only in the colons of animals that received DMH, whose results are presented in Figure 7. No significant differences were observed in the ACF/AC ratio between the treatment groups, which ranged from 1.1 to 2.0. The BFRP treatments led to reductions in the frequencies of the preneoplastic lesions induced by the carcinogen DMH. The differences were statistically significant for the group of animals treated with 6 mg/kg of the BFRP when compared to the group treated with DMH alone, corresponding to a 41.6% reduction in preneoplastic lesions.

DMH is an alkylating procarcinogen that requires metabolic activation, involving various hepatic enzymes [42]. In the liver, it is metabolized into a diazonium ion, causing an imbalance in oxidative stress and DNA damage and contributing to carcinogenesis, including the development of ACF [43]. Processes related to inflammation have been demonstrated to play a role in the development of colon carcinogenesis, both in humans and in cases induced by DMH [44]. In this context, the reduction in preneoplastic lesions induced by DMH is related, at least in part, to the anti-inflammatory property of BFRP.

It is noteworthy that the literature has reported the anti-inflammatory activity of the RHPE, from which the BFRP was developed [11,12,13,15]. In addition, our previous study demonstrated that the animals treated with DMH showed a high expression of cyclooxygenase-2 (COX-2) in the colon tissue. RPHE significantly decreased the expression induced by DMH, indicating its anti-inflammatory properties [13]. COX-2 is an enzyme crucial in inducing inflammatory responses and is often associated with colorectal tumors. Under the same experimental conditions, we also observed the chemopreventive effect of the hydroalcoholic extract of *D. ecastaphyllum* stems, the botanical source of red propolis responsible for providing phenolic compounds. The treatments with this extract led to a reduction in DMH-induced preneoplastic lesions, which was accompanied by significantly lower levels of COX-2 expression [16]. Taken together, it is noted that the BFRP exhibited a protective effect against colon carcinogenesis at a dose lower (6 mg/kg) than that observed for the RPHE (12 mg/kg) and its botanical source (48 mg/kg).

Significant differences were observed in the weight gain of animals treated with DMH and DMSO plus DMH in relation to the control vehicle. In relation to water consumption, there were no statistically significant changes when compared to the vehicle control group (Table 4).

Previous studies have documented a notable reduction in body weight in rats treated with DMH when compared to the control group. This decline in body weight could be attributed to alterations in fat metabolism resulting from increased gluconeogenesis to meet the heightened energy demands of the hyperplastic cells [13,45]. Although there were no significant changes in food intake, the reduction in body weight observed after DMH treatment may be associated with the overall breakdown of lipids and proteins [46]. In contrast, the animals that received BFRP along with DMH did not exhibit significant decreases in body weight gain when compared to those that were not treated with DMH. These data indicate that the SPRE exhibited a protective effect against DMH-induced toxicity. Therefore, the BFRP was able to reduce not only the preneoplastic lesions induced by DMH but also its toxicity.

The complex chemical composition of an extract, containing various compounds, presents a notable challenge in understanding its mechanisms of action. These constituents can produce a range of biological effects, either working together or independently. Despite this complexity, an extract’s biological activity is primarily tied to the synergistic effects of its constituents influencing various levels, targets, and pathways simultaneously. Thus, the extracts may demonstrate biological properties not observed in the isolated constituents [47,48].

Epidemiological studies show that flavonoid-rich foods prevent some diseases, including metabolic-related diseases and cancer. Further studies show that flavonoids have many properties, which include antioxidant, anti-inflammatory, anti-proliferative [49,50]. These results underscore the relevance of BFRP, an extract rich in flavonoids, as a potentially beneficial intervention in inflammatory bowel conditions [51].

## 3. Material and Methods

### 3.1. Obtaining the Hydroalcoholic Extract of Red Propolis (RPHE) and the Benzophenone-Free Red Propolis Extract (BFRP)

The Brazilian red propolis sample was sourced in February 2020 from the Association of Beekeepers of Canavieiras (COAPER, Bahia, Brazil), a cooperative situated in the southern region of Bahia, northeastern Brazil. The preparation of RPHE followed the method described by Aldana-Mejía et al. [21].

To prepare the BFRP, 50 g of crude red propolis was frozen at −80 °C for 3 h, then ground using a mill. The ground material was subjected to dynamic maceration in 500 mL of n-hexane at 35 °C and 140 rpm. The solvent was replaced every 24 h, completing three maceration cycles. After that, the precipitate was submitted to another three-times extraction using 300 mL of cold methanol (10 °C) for 30 min to give 22.7 g, a 45.4% yield of BFRP after concentration in a Buchi^®^ rotary evaporator apparatus.

### 3.2. Isolation and Identification of the Standard Compounds

Twelve grams of BFRP was subjected to silica gel column chromatography (120 × 5 cm) with hexane/ethyl acetate in a gradient mode from 100% of hexane to 50% of hexane. This fractionation resulted in 247 fractions of 8 mL each that were analyzed by Thin Layer Chromatography (TLC) (hexane/ethyl acetate, 70:30) and reunited to give 26 fractions (F 1-26). Fraction 4 was submitted to a preparative scale using a high-performance liquid chromatography (HPLC; Shimadzu, Kyoto, Japan) with water plus 0.1% acetic acid (solvent A) and acetonitrile (solvent B). The separation method was a gradient starting in 30% of solvent B to 100% of solvent B in 25 min, furnishing isoliquiritigenin (**1**) (41 mg). Fraction 8 formed a precipitate, identified as vestitol (**2**) (1388 mg). Fraction 12 (1.7 g) was submitted to Sephadex column chromatography (90 × 3 cm) with methanol/water to give neovestitol (**3**) (33 mg) and medicarpin (**4**) (18 mg). Fraction 17 (685 mg) was submitted to a preparative HPLC with water plus 0.1% glacial acetic acid (solvent A) and acetonitrile (solvent B). The separation method was a gradient starting in 45% of solvent B to 100% of solvent B in 30 min, furnishing 7-*O*-methylvestitol (**5**) (41 mg).

The isolated compounds were identified through a nuclear magnetic resonance (NMR) analysis for the structural elucidation. The NMR spectra were obtained on a Bruker Advance DRX400, operating at 400 and 100 mHz for ^1^H and ^13^C, respectively. Deuterated solvent from Sigma Aldrich^®^ was used.

### 3.3. RP-HPLC-PDA Method for BFRP Analysis

The analytical method for BFRP was developed using a Shimadzu HPLC system, model LC-20AR Prominence, equipped with an SIL-10AF autosampler, CTO-20A column oven, CBM-20A communication bus module, DGU-20A3R in-line degasser, and an SPD-M20A photodiode array detector. All experiments were conducted in triplicate under controlled temperature conditions. The HPLC analysis employed a Shim-pack VP-ODS analytical column (250 × 4.6 mm i.d., 5 μm; Shimadzu), with acidified water (0.2% acetic acid) as solvent A and acetonitrile as solvent B, using the following parameters: 0.01−15.00 min, 30−48% of B; 15.00−20.00 min, 48−55% of B; 20.00−23.00 min, 55−73% of B; 23.00−24.00 min, 73−100% of B; 24.00−29.00 min, 100% of B; 29.00−30.00 min, 100−40% of B; and 30.00−32.00 min, 40% of B. The flow rate was set at 1 mL/min and the UV detection at 280 nm for the compounds. The temperature of the column was set at 40 °C, and the injected volume was 20 μL. The Lab solution^®^ software was used to process the data.

### 3.4. Validation of the Method

The validation followed the guidelines set by the Brazilian National Health Surveillance Agency (ANVISA) [23]. Parameters such as selectivity, linearity, detection and quantification limits, accuracy, precision, recovery, and robustness were assessed. All validation experiments were conducted in triplicate. 

#### 3.4.1. Selectivity

Method selectivity was determined based on the separation efficiency of chromatographic peaks, evaluated by assessing the chromatographic resolution. The peaks were compared to authentic standards to confirm the selectivity of red propolis compounds, using parameters such as relative retention time, peak area, resolution, and UV spectra.

#### 3.4.2. Linearity, Limit of Detection, and Limit of Quantification

The linearity was estimated by calibration curves. The linear dynamic range was performed for the concentrations of 600, 500, 400, 300, 200, 100, 75, 25, and 15 μg/mL from the analytes. 1,2,4,5- tetramethylbenzene (TMB) was used as the internal standard at a final concentration of 100 μg/mL. TMB was selected as the internal standard due to its chemical stability, and non-interference with their detection, ensuring reliable chromatographic performance. From the solutions, 20 μL were injected in triplicate for three consecutive days. The analytical curve was generated by plotting the ratio of each analyte standard to the internal standard areas. Linearity was assessed by calculating the correlation coefficient (R), determination coefficient (R²), and performing the lack of fit test. The Statistic 8.0 software was used to determine the minimum and maximum observed residual values. Limits of detection (LD) and quantification (LQ) were calculated based on baseline noise, using the following equations: LD = (3.3 × SD) / IC and LQ = (10 × SD) / IC, where SD represents the standard deviation of the y-intercept when x equals zero from the calibration curves, and IC is the slope of the analytical curve [23].

#### 3.4.3. Precision and Accuracy

Three concentrations were selected to evaluate the precision and accuracy of the method. The method’s precision was evaluated by calculating the relative standard deviation (RSD) for repeatability at three concentration levels: high (400 μg/mL), medium (100 μg/mL), and low (25 μg/mL). Measurements were taken on the same day (intraday precision) and across three consecutive days (interday precision). Accuracy was determined by comparing the theoretical and actual values of the solutions at the same three concentrations.

#### 3.4.4. Recovery

For the recovery assessment, 15 g of ground propolis matrix underwent exhaustive extraction in a Soxhlet apparatus for 8 h with 300 mL of 96% ethanol. The matrix was then dried in an air-circulating oven at 50 ºC for 12 h, and the extract was concentrated under vacuum. A standard solution of the compounds was prepared at a final concentration of 100 μg/mL for each compound. Following this, 200 mg of propolis matrix was spiked with the standard solution at three concentration levels: 4 mL (400 μg/mL, high level), 1 mL (100 μg/mL, medium level), and 0.25 mL (25 μg/mL, low level), and dried at room temperature. For extraction, 70% ethanol was used, with TMB (100 μg/mL) as the internal standard, and benzophenone (100 μg/mL) as the secondary internal standard. The extraction process was carried out using a shaker incubator at 35 ºC and 140 rpm for 120 min. The standard compounds were quantified using the developed HPLC method. All experiments were performed in triplicate, and recovery percentages were calculated by comparing the theoretical and actual concentration values.

#### 3.4.5. Robustness

The robustness was determined with the Plackett–Burman design for seven factors and eight experiments that combined the higher (+1) and lower (−1) levels of each factor. The factors were the flow rate (0.9 and 1.1 mL/min), the oven temperature (35 and 45 °C), the percentage of organic solvent (+1 and −1%), the detection wavelength (271 and 281 nm), and the volume of injection (19 and 21 μL). These factors were slightly changed for the higher and lower levels. To calculate the effects, the following equation was used: $Ex=\frac{\sum y\left(+\right)-\sum y(-)}{n/2}$, where Ex is the estimated effect of the response; Σy(+) is the sum of the responses at the positive level; Σy(−) is the sum of the responses at the negative level; and *n* is the number of experiments of the experimental design. These were expressed as the variation coefficients in percentages. The concentrations used to evaluate the robustness were the low, medium, and high levels, which were the same as were used in the precision test.

### 3.5. Toxicity Assessment

#### 3.5.1. Fish Toxicity Test

##### Zebrafish Husbandry

Adult zebrafish (*Danio rerio*, AB2O2 line, Karlsruhe Institute of Technology; Eggenstein-Leopoldshafen, Germany) were bred and reared according to the standard methods [52]. The adult fish were kept under a light-dark cycle of 14 h of light and 10 h of darkness at a temper-ature of 28 °C. Fertilized eggs were gathered within 2 h of being laid and placed in petri dishes (10 cm in diameter) filled with E3 medium (5 mM NaCl, 0.17 mM KCl, 0.33 mM CaCl_2_) for incubation at 28.5 °C. The eggs from different clutches were kept separately to carry out the experiments, and all the experiments were performed in triplicate. All zebrafish care and procedures were carried out in accordance with the animal welfare regulations in Germany and received approval from the Government of Baden-Württemberg, Regierungspräsidium Karlsruhe, Germany (reference number 35-9185.64/BH KIT/March 1, 2022).

##### Vibration Startle Response Assay

A vibration startle response (VSRA) assay was conducted on 5 days postfertilization (dpf) larvae, after exposure to seven concentrations of BFRP (2.5, 5, 10, 20, 40, 80, and 160 µg/mL) for 48 h. In parallel, assays were conducted with the red propolis hydroalcoholic extract (RPHE), under the same experimental conditions, aiming to compare the effects. Ten larvae 72 h postfertilization (hpf) per condition were distributed in 5 cm diameter Petri dishes. The E3 medium was completely removed and replaced with 9 mL of E3 and 1 mL of the BFRP or RPHE solution. The exposure concentrations, based on previous studies, ranged from 2.5 to 160 µg/mL [17]. A stock solution of the BFRP and RPHE at 50 mg/mL was prepared, diluted in a mixture composed of 50% dimethyl sulfoxide (DMSO; Sigma-Aldrich) and 50% E3 medium. The DMSO concentration in the plate at the highest evaluated concentration of the BFRP and RPHE (160 µg/mL) was equivalent to 0.1%. Negative (E3 medium) and positive (10% tricaine) controls were added to the experiment [53].

The VSRA assay was based on the automated delivery of vibrational stimuli. The larvae were subjected to vibration at 5 dpf using sound-based vibration equipment, which was developed by the Screening Center at the Karlsruhe Institute of Technology and the Institute of Toxicology and Genetics (Karlsruhe, BW, Germany). The videos recorded by the equipment were manually analyzed, and the number of larvae that exhibited a ‘c-shape’, as well as those with little movement and/or that were immobile, was assessed as described in Hayot et al. [53].

##### Morphological Analyses

The larvae at 4.5 dpf were treated with 40 µg/mL of the BFRP and RPHE for 4 h, and the larval behavior before death was assessed. The choice of the concentration and treatment time was based on the results of the VSRA test. In this assay, the BFRP exhibited toxicity starting at 40 µg/mL. The RPHE was treated at the same concentration for comparison. The larvae were imaged using a *brightfield and* fluorescence microscope (with an excitation wavelength of 395 nm and an emission wavelength of 510 nm) Nikon SMZ18, with a camera Nikon Digital Sight 10, on a lateral view. The body thickness was measured as a toxicity parameter, as well as the swim bladder and otic vesicle area. Three replicates of ten larvae were analyzed. The parameters were measured using ImageJ, and the averages of the obtained values were calculated.

#### 3.5.2. Genetic Toxicology Test

The peripheral blood micronucleus assay was utilized to evaluate the genotoxicity of BFRP and was carried out in accordance with the guidelines outlined in OECD 474 [37] recommendations. Male Swiss mice (*Mus musculus*), each weighing around 30 g, were sourced from the Faculty of Pharmaceutical Sciences at the University of São Paulo (Ribeirão Preto, SP, Brazil). The study protocols received approval from the Animal Use Ethics Committee of the University of Franca (process number 9907081122).

Following a one-week acclimatization period, the rodents were assigned into groups of five. Three distinct doses of BFRP were tested: 500, 1000, and 2000 mg/kg. The extract was administered to the animals via gavage over two consecutive days, once per day. Control groups included a solvent (DMSO, 1%) and a positive control group (doxorubicin; Bergamo, Taboão da Serra, SP, Brazil; 10 mg/kg). Peripheral blood samples were collected from the tail vein 48 h after the final treatment. Euthanasia of the rodents was performed through the intraperitoneal injection of thiopental sodium (840 mg/kg, Cristália, Itapira, SP, Brazil).

Blood samples were obtained through cardiac puncture under anesthesia (sodium pentobarbital, 45 mg/kg, intraperitoneal, 0.3 mL) for biochemical toxicity analysis. Serum levels of alanine aminotransferase (ALT) and creatinine were measured to assess hepatotoxicity and nephrotoxicity, respectively. For this, 0.3 mL serum samples were prepared using biochemical reagent kits Creatinine K (ref. 96) and ALT Liquiform (ref. 108) from Labtest Diagnóstico (Lagoa Santa, MG, Brazil), following the manufacturer’s instructions. Measurements were conducted using a microprocessor-controlled random access biochemical analyzer (Bioplus, Barueri, SP, Brazil).

The frequency of micronucleated polychromatic erythrocytes (MNPCEs) was evaluated by examining 5000 polychromatic erythrocytes (PCEs) per mouse using a bright-field microscope (immersion objective, ×1000). Cytotoxicity resulting from the treatments was determined by calculating the ratio of PCE to the total erythrocyte count (PCE + normochromatic erythrocytes), analyzing 2000 erythrocytes per mouse. 

### 3.6. Evaluation of the Effect of BFRP on Preneoplastic Colon Lesions

The aberrant crypt foci (ACF) assay was conducted following the methodology established by Bird [54] to assess the anticarcinogenic potential of BFRP. Male Wistar Hannover rats (Rattus norvegicus), each weighing approximately 120 g, were supplied by the Faculty of Pharmaceutical Sciences at the University of São Paulo (Ribeirão Preto, SP, Brazil) for the experiments. The animals (five animals/group) were housed in plastic cages in an experimental facility with controlled temperature (23 ± 2 °C) and humidity (50 ± 10%) under a 12-h light/dark cycle, with unrestricted access to food and water. The treatment protocols received approval from the Ethics Committee on Animal Use of the University of Franca (process number 9907081122).

The doses of the extract evaluated (6, 12, 24, and 48 mg/kg) were established based on a previous study involving red propolis hydroalcoholic extract [13]. The carcinogen 1,2-dimethylhydrazine (DMH; Sigma-Aldrich; 40 mg/kg) was utilized to induce preneoplastic lesions, specifically ACF, in the colon of the rats. This carcinogen was dissolved immediately prior to administration in 1 mM ethylenediaminetet-raacetic acid (EDTA; Labsynth, Diadema, SP, Brazil).

Following a week of acclimatization, various doses of BFRP were administered daily, once per day, for two weeks (during the second and third weeks) via gavage (1.0 mL/animal). Concurrently, DMH was administered subcutaneously twice a week (on the second and fifth days) for two weeks, totaling four administrations. Control groups included solvent (DMSO, 1%), DMH (40 mg/kg), and a combination of solvent and DMH. For the subsequent two weeks, the animals received no treatment. Body weight and water intake were recorded three times a week throughout the experimental period. The procedures for colon removal, slide preparation, and analysis were performed in accordance with Furtado et al. [55]. The frequencies of ACF and aberrant crypts (AC) were assessed by examining 15 sequential fields per colon segment.

### 3.7. Statistical Analysis

The data obtained in animals, zebrafish and rodents, were analyzed using analysis of variance (ANOVA) for fully randomized experiments, where F statistics and corresponding p-values were calculated. For results where *p* < 0.05, treatment means were compared using Tukey’s test, and the minimum significant difference was determined for α = 0.05. The statistical analysis was performed using Graph Pad Prism program 6. The percentage reduction in preneoplastic lesions was calculated according to Furtado et al. [55].

## 4. Conclusions

A benzophenones-free red propolis extract was obtained by a high-performance liquid chromatography with photodiode array detection method that is in accordance with the validation guidelines of the Brazilian regulatory agency. Toxicological evaluation assays revealed that benzophenones-free red propolis extract was significantly less toxic than hydroalcoholic extract of red propolis in zebrafish larvae and showed no genotoxicity in rodents. Therefore, it is confirmed that the removal of polyprenylated benzophenones from Brazilian red propolis leads to a reduction in the toxicity of this beekeeping product. Furthermore, benzophenones-free red propolis extract has promising effect on preventing colon cancer. How-ever, future research should focus on investigating the long-term effects of this extract in larger animal models and human clinical trials to further validate its safety and efficacy. Additionally, exploring the underlying mechanisms by which the extract exerts its chemopreventive effects is essential for fully understanding its therapeutic potential.

## Figures and Tables

**Figure 1 pharmaceuticals-17-01340-f001:**
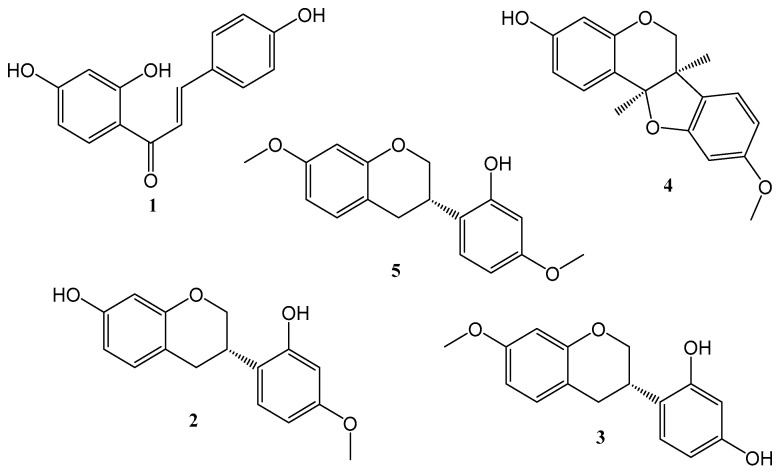
Chemical structures of isolated compounds from BFRP: **1**—isoliquiritigenin; **2**—vestitol; **3**—neovestitol; **4**—medicarpin; **5**—7-*O*-methylvestitol.

**Figure 2 pharmaceuticals-17-01340-f002:**
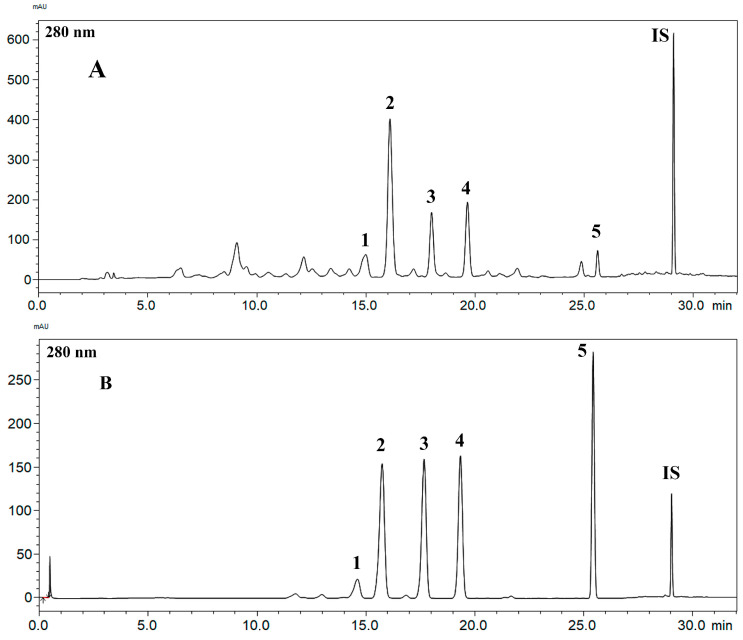
RP-HPLC-PDA chromatogram of the BFRP (**A**) in comparison with the respective standards (**B**): **1**—isoliquiritigenin; **2**—vestitol; **3**—neovestitol; **4**—medicarpin; **5**—7-*O*-methylvestitol; and IS: benzophenone.

**Figure 3 pharmaceuticals-17-01340-f003:**
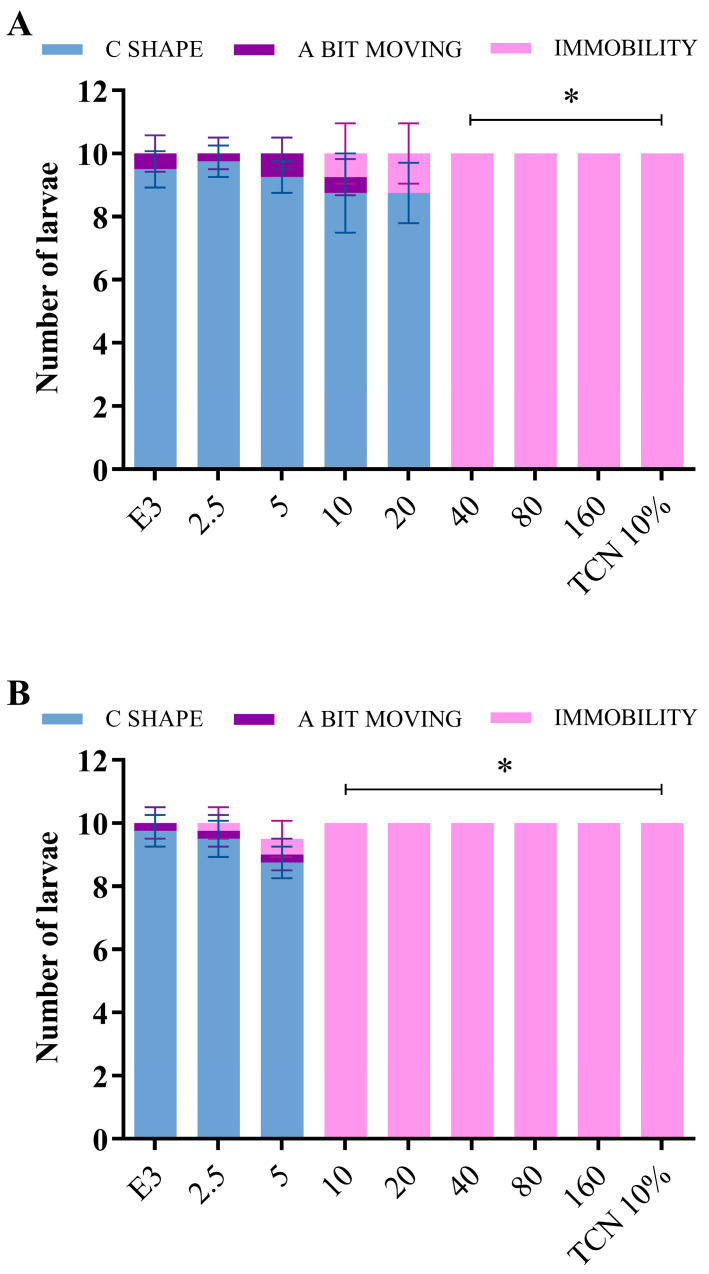
Effects of different concentrations of BFRP (**A**) and RPHE (**B**) on the startle response of zebrafish larvae (5 dpf) with VSRA assay. The count included the number of larvae that were immobile, a bit moving, and performing the C shape at the time of the stimulus. E3—E3 medium (negative control); BFRP—benzophenone-free red propolis extract; RPHE—red propolis hydroalcoholic extract; and TCN—tricaine 10% (positive control). The values are the mean ± SD (*n* = 10). The experiment was performed in triplicate. * significantly different from the negative control (E3) group (*p* < 0.0001).

**Figure 4 pharmaceuticals-17-01340-f004:**
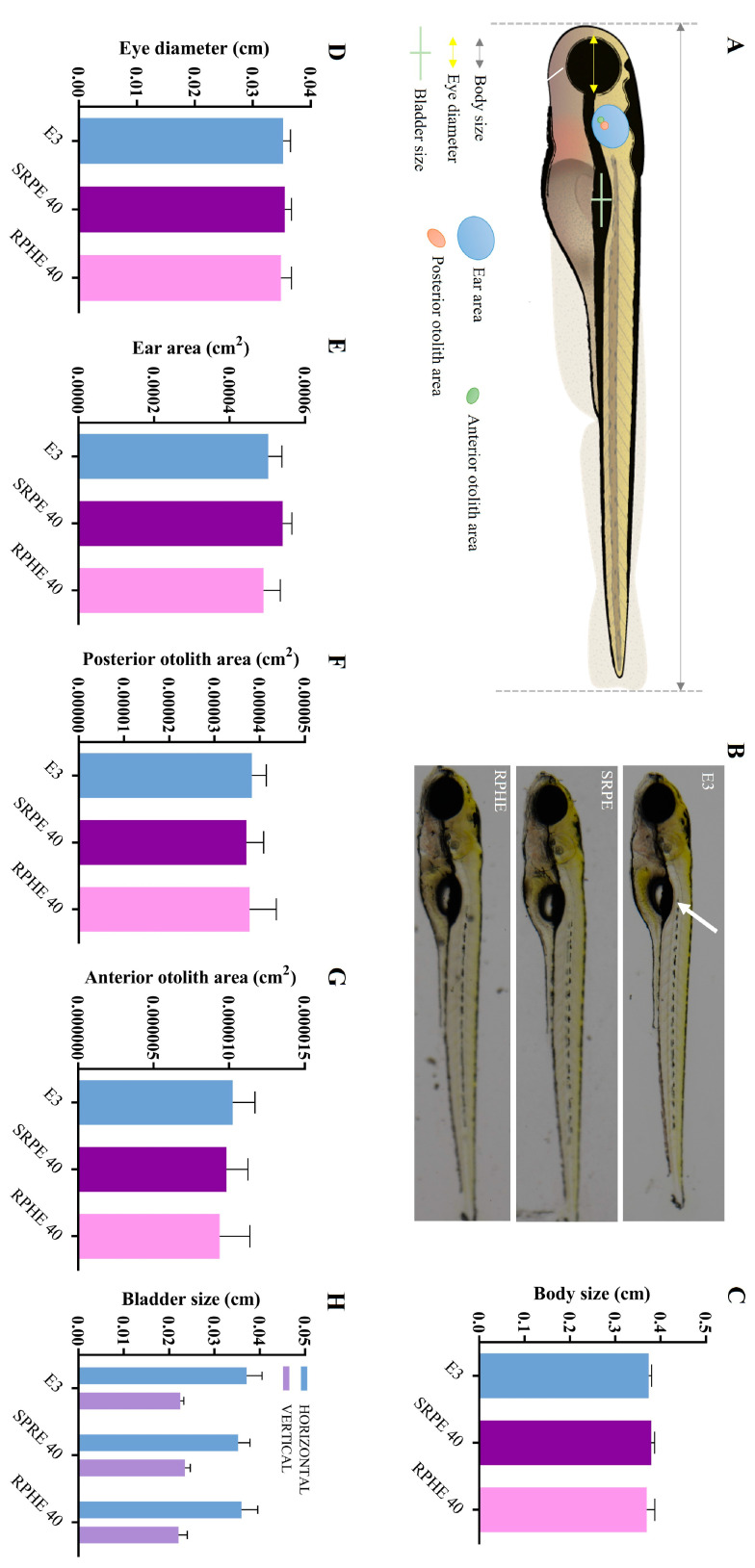
Effects of BFRP and RPHE on the morphology of zebrafish larvae at 5 dpf after 4 h of exposure. (**A**) Schematic representation of the different measurements of the larva (modified from Crouzier et al. [27]). (**B**) Lateral view of 5 dpf larvae. Swim bladder is indicated by a white arrow. (**C**) Measurement of body length, from the mouth to the end of the tail fin. (**D**) Measurement of the diameter of the eye. (**E**–**G**) Measurement of the ear. (**H**) Measurement of swim bladder size. BFRP—benzophenone-free red propolis extract (40 µg/mL); and RPHE—red propolis hydroalcoholic extract (40 µg/mL). No significant difference was observed between treated and untreated larvae (*p* > 0.05). The values are the mean ± SD (*n* = 10). Scale = 200 µm (75×) with a *brightfield microscope* Nikon SMZ18, with a camera Nikon Digital Sight 10.

**Figure 5 pharmaceuticals-17-01340-f005:**
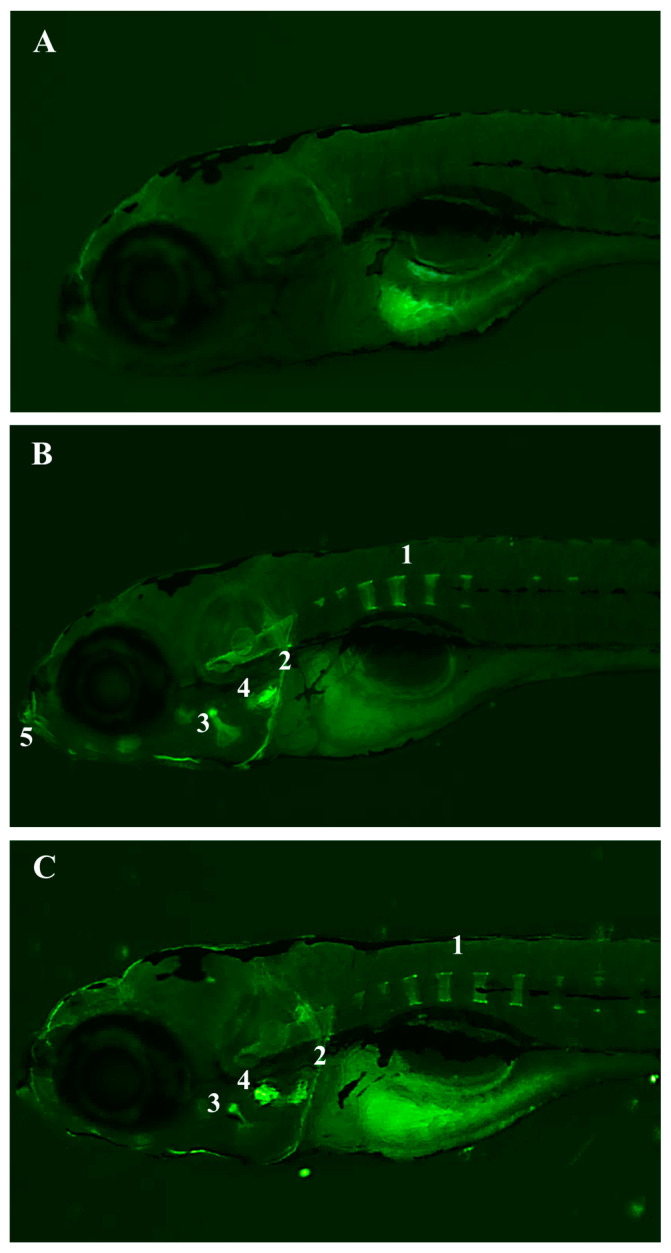
Effects of BFRP and RPHE on the development of zebrafish after 4 h of exposure. (**A**) Lateral view of 5 dpf zebrafish larvae treated with E3 medium (negative control). (**B**) Lateral view of 5 dpf zebrafish larvae treated with BFRP. (**C**) Lateral view of 5 dpf zebrafish larvae treated with RPHE. BFRP—benzophenone-free red propolis extract (40 µg/mL); RPHE—red propolis hydroalcoholic extract (40 µg/mL); 1—notochord; 2—branchial arch; 3 and 4—unrecognized structures; and 5—mouth. Fluorescence microscope (excitation wavelength 395 nm; emission wavelength 510 nm) Nikon SMZ18, with a camera Nikon Digital Sight 10. Scale = 200 µm (82×).

**Figure 6 pharmaceuticals-17-01340-f006:**
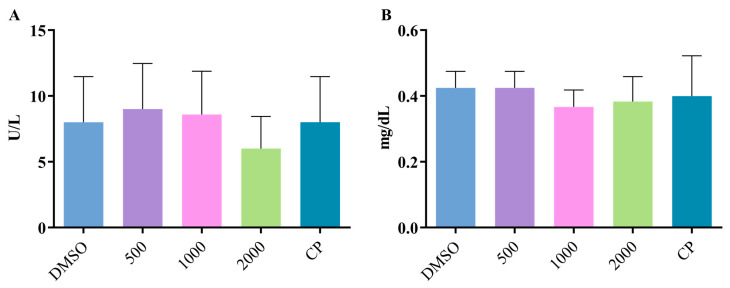
Serum levels of ALT (**A**) and creatinine (**B**) obtained in the peripheral blood of Swiss mice treated with different doses of BFRP (500, 1000, and 2000 mg/kg) and their respective controls. BFRP—benzophenone-free red propolis extract; ALT—alanine aminotransferase; DMSO—dimethyl sulfoxide (5%); and CP—positive control (doxorubicin, 10 mg/kg). The values are the mean ± SD (*n* = 5).

**Figure 7 pharmaceuticals-17-01340-f007:**
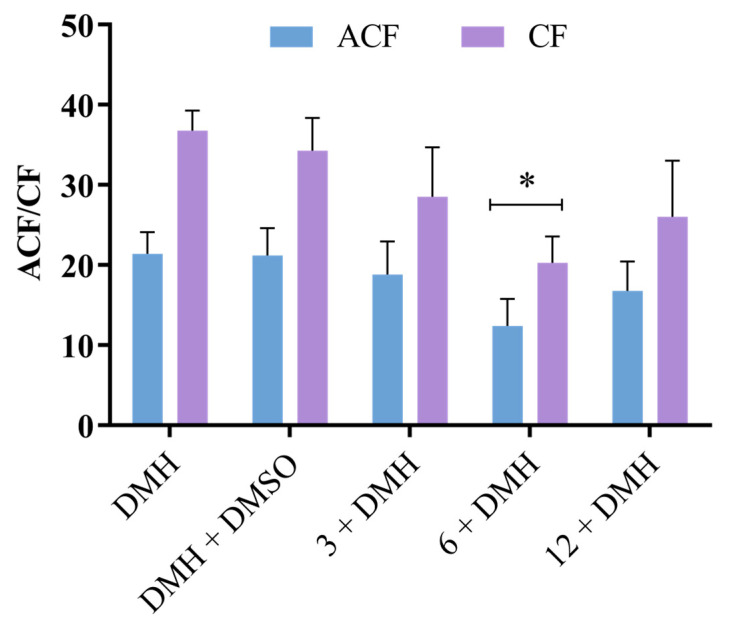
Effect of BFRP on DMH-induced carcinogenesis in the colon of Wistar Hannover rats. Rates of aberrant crypt foci (ACF) and aberrant crypts (AC) identified in the distal colon of rats administered BFRP and DMH. BFRP—benzophenone-free red propolis extract (3, 6, and 12 mg/kg); ACF—aberrant crypt foci; AC—aberrant crypt; DMH—1,2-dimethylhydrazine (40 mg/kg); and DMSO—dimethyl sulfoxide (5%). The values are the mean ± SD (*n* = 5). * significantly different from the DMH group (*p* < 0.05).

**Table 1 pharmaceuticals-17-01340-t001:** Linearity, limits of detection, and quantification of the method.

Compound	Equation	R^2^	R	LD	LQ	Minimum Observed Residual Value	Maximum Observed Residual Value	Lack of Fit *p* Value
(1)	y = 0.006x + 0.063	0.998	0.998	3.94	10.99	−3.31823	3.23520	0.61
(2)	y = 0.0356x + 0.3835	0.999	0.999	2.90	8.96	−2.2265	2.2288	0.79
(3)	y = 0.0313x + 0.4228	0.998	0.998	2.78	9.42	−2.26225	1.62954	0.74
(4)	y = 0.0293x + 0.4153	0.997	0.998	2.62	9.15	−2.26638	1.84199	0.81
(5)	y = 0.0292x + 0.4808	0.998	0.998	1.31	3.97	−1.59636	1.26722	0.93

R^2^—determination coefficient; R—correlation coefficient; LD—limit of detection (µg/mL); LQ—limit of quantification (µg/mL); **1**—isoliquiritigenin; **2**—vestitol; **3**—neovestitol; **4**—medicarpin; and **5**—7-*O*-methylvestitol.

**Table 2 pharmaceuticals-17-01340-t002:** Precision and accuracy of the method.

Compound	Level	Precision (RSD)		Accuracy (%)	E (%)	Recovery (%)
		Intraday	Interday			
(1)	High	0.08	2.13	100.89 ± 2.7	−0.89	104.02 ± 1.2
	Medium	0.79	1.38	113.93 ± 1.1	3.93	101.10 ± 2.8
(2)	Low	0.66	2.97	97.93 ± 0.8	4.07	99.26 ± 1.9
	High	0.35	1.65	98.49 ± 1.9	−1.51	107.02 ± 0.7
	Medium	0.65	2.99	107.7 ± 10.2	4.71	106.55 ± 1.1
(3)	Low	0.46	1.15	104.04 ± 1.1	3.96	96.67 ± 0.9
	High	0.57	0.72	99.50 ± 0.2	−0.50	108.69 ± 1.7
	Medium	0.61	0.44	110.92 ± 0.3	−2.92	105.01 ± 1.6
(4)	Low	0.39	1.25	96.05 ± 0.9	3.59	99.50 ± 2.1
	High	0.51	0.76	106.38 ± 1.2	2.38	97.52 ± 1.3
	Medium	0.56	0.43	108.05 ± 0.8	−1.35	100.94 ± 2.4
(5)	Low	1.36	0.37	106.50 ± 1.2	2.65	102.86 ± 1.9
	High	0.61	2.63	97.89 ± 2.5	2.11	99.65 ± 0.4
	Medium	0.44	0.43	101.94 ± 1.7	−1.94	96.90 ± 1.1
I.S	Low	0.43	0.97	98.94 ± 2.3	−1.06	100.23 ± 2.2
	High					100.98 ± 1.2
	Medium					96.71 ± 0.9
	Low					100.39 ± 1.6

RSD—relative standard deviation (%); E—error; **1**—isoliquiritigenin; **2**—vestitol; **3**—neovestitol; **4**—medicarpin; **5**—7-*O*-methylvestitol; and IS—internal standard, benzophenone.

**Table 3 pharmaceuticals-17-01340-t003:** Frequencies of micronucleated polychromatic erythrocytes (MNPCEs) in the peripheral blood of mice treated with different doses of BFRP.

Treatments (mg/kg)	MNPCEs	PCE/PCE + NCE
**DMSO**	6.0 ± 2.0	0.04 ± 0.02
**500**	5.0 ± 1.9	0.04 ± 0.01
**1000**	7.8 ± 2.6	0.04 ± 0.01
**2000**	9.2 ± 1.3	0.05 ± 0.00
**PC**	30.3 ± 3.2 *	0.03 ± 0.01

BFRP—benzophenone-free red propolis extract; PCEs—polychromatic erythrocytes; NCEs—normochromatic erythrocytes; DMSO—dimethyl sulfoxide (5%); and PC—positive control (doxorubicin, 10 mg/kg). The values are the mean ± SD (*n* = 5). * significantly different from the negative control group (*p* < 0.05).

**Table 4 pharmaceuticals-17-01340-t004:** Average initial body weight, final body weight, body weight gain, and water consumption of rats treated with varying doses of BFRP alongside DMH, along with their corresponding control groups, during the four-week experimental period.

Treatments (mg/kg)	Initial Weight (g)	Final Weight (g)	Weight Gain (g)	Water Consumption (mL/Animal/Day)
**DMSO**	225.8 ± 5.2	352.8 ± 2.7	128.0 ± 5.3	53.9 ± 18.1
**24**	229.3 ± 4.0	350.3 ± 5.5	121.0 ± 4.6	50.7 ± 12.1
**DMH**	227.0 ± 7.0	296.0 ± 9.5	89.8 ± 10.4 *	65.9 ± 22.2
**DMSO + DMH**	230.6 ± 7.0	295.7 ± 8.1	76.8 ± 14.0 *	49.0 ± 12.6
**3 + DMH**	219.8 ± 6.6	340.8 ± 6.2	122.2 ± 7.7	63.2 ± 12.5
**6 + DMH**	218.8 ± 5.5	341.7 ± 9.8	121.4 ± 6.9	52.2 ± 13.3
**12 + DMH**	225.8 ± 6.9	348.2 ± 7.2	122.4 ± 8.4	73.6 ± 23.3

BFRP—benzophenone-free red propolis extract; DMH—1,2-dimethylhydrazine (40 mg/kg); and DMSO—dimethyl sulfoxide (5%). The values are the mean ± SD (*n* = 5). * significantly different from the DMSO group (*p* < 0.05).

## Data Availability

Data is contained within the article.

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
