# Peer review of "Development of a Benzophenone-Free Red Propolis Extract and Evaluation of Its Efficacy against Colon Carcinogenesis"

_pharmaceuticals, 2024, doi:10.3390/ph17101340_

Round 1

Reviewer 1 Report

Comments and Suggestions for Authors

The authors extracted a benzophenones-free red propolis extract (BFRP) and develop new HPLC-PDA method for assay the main constituents. Besides, The toxic potential of BFRP was evaluated using zebra fish larvae in a vibrational startle response assay (VSRA). However, minor revisions are required.

1- The title should be modified to reflect the less toxicity of red propolis extract   or a benzophenones-free red propolis extract

2- Figure 3 should be colured and with adjusted space between A and B.

3- Figure 4 c is not seen. Figures 4 and 6 should be colured also.

4- Credit author statements should follow journal guidelines

e.g. Author Contributions

x and y.: methodology, funding acquisition, project administration, and editing. z., A., B and C.: supervision, resources, writing—review and editing.

5- The following articles should be cited in the introduction

6- Review article

Red propolis: Chemical composition and pharmacological activity

https://www.sciencedirect.com/science/article/pii/S2221169117305269

a Silva LH, Squarisi IS, de Freitas KS, Barcelos Ribeiro A, Ozelin SD, Aldana-Mejía JA, de Oliveira LT, Rodrigues TE, de Melo MR, Nicolella HD, Alves BS. Toxicological and chemoprevention studies of Dalbergia ecastaphyllum (L.) Taub. stem, the botanical source of Brazilian red propolis. Journal of Pharmacy and Pharmacology. 2022 May

Aldana-Mejia JA, Ccana-Ccapatinta GV, Squarisi IS, Nascimento S, Tanimoto MH, Ribeiro VP, Arruda C, Nicolella H, Esperandim T, Ribeiro AB, de Freitas KS. Nonclinical toxicological studies of Brazilian red propolis and its primary botanical source Dalbergia ecastaphyllum. Chemical Research in Toxicology. 2021 Mar 15;34(4):1024-33.

de Mendonça IC, Porto IC, do Nascimento TG, de Souza NS, Oliveira JM, Arruda RE, Mousinho KC, dos Santos AF, Basílio-Júnior ID, Parolia A, Barreto FS. Brazilian red propolis: phytochemical screening, antioxidant activity and effect against cancer cells. BMC Complementary and Alternative Medicine. 2015 Dec;15:1-2.

7- In line 88, add missed solvent (hexane: 70:30) and

8- The font size and style in lines 103 to 106 should be the aligned with all the text.

9- Why tetramethylbenzene (TMB) was used as internal standard ? mention the reasons in section 2.4.2

10- In line 181, the date for ethical approval should reported.

11- Write full names for all abbreviations when first mentioning e.g. EDTA, DMH

12- In depth discussions with comparisons with other types of propolis are strongly recommended ?

13-  Abbreviation list is strongly recommended

14- Add future plans and study limitation as one paragraph at the end of the discussion or the conclusion

15- Conclusion should be written with full names for all abbreviations HPLC-PDA, BFRP, and RPHE

Author Response

We would like to thank the Reviewer for all the insightful and constructive comments on our work. Each comment is addressed below in a point-by-point way, as well as all changes made on the manuscript are highlighted in yellow for better understanding.

The authors extracted a benzophenones-free red propolis extract (BFRP) and develop new HPLC-PDA method for assay the main constituents. Besides, the toxic potential of BFRP was evaluated using zebra fish larvae in a vibrational startle response assay (VSRA). However, minor revisions are required.

1- The title should be modified to reflect the less toxicity of red propolis extract or a benzophenones-free red propolis extract.

Answer: The title of the manuscript has been changed as suggested.

"Development of a benzophenones-free red propolis extract and evaluation of its efficacy against colon carcinogenesis."

2- Figure 3 should be colored and with adjusted space between A and B.

Answer: Figure 3 was presented in color and the spacing between panels A and B has been adjusted as requested (page 21).

3- Figure 4 c is not seen. Figures 4 and 6 should be colored also.

Answer: Figure 4 C has been made visible (pages 23 and 24). Figures 4 and 6 were presented in color as requested (pages 23 and 24, 28).

4- Credit author statements should follow journal guidelines

e.g. Author Contributions

x and y.: methodology, funding acquisition, project administration, and editing. z., A., B and C.: supervision, resources, writing—review and editing.

Answer: The author’s contribution statements have been revised to comply with the journal's guidelines as requested (page 34, lines 653-658).

5-The following articles should be cited in the introduction

Red propolis: Chemical composition and pharmacological activity

https://www.sciencedirect.com/science/article/pii/S2221169117305269

a Silva LH, Squarisi IS, de Freitas KS, Barcelos Ribeiro A, Ozelin SD, Aldana-Mejía JA, de Oliveira LT, Rodrigues TE, de Melo MR, Nicolella HD, Alves BS. Toxicological and chemoprevention studies of Dalbergia ecastaphyllum (L.) Taub. stem, the botanical source of Brazilian red propolis. Journal of Pharmacy and Pharmacology. 2022 May

Aldana-Mejia JA, Ccana-Ccapatinta GV, Squarisi IS, Nascimento S, Tanimoto MH, Ribeiro VP, Arruda C, Nicolella H, Esperandim T, Ribeiro AB, de Freitas KS. Nonclinical toxicological studies of Brazilian red propolis and its primary botanical source Dalbergia ecastaphyllum. Chemical Research in Toxicology. 2021 Mar 15;34(4):1024-33.

de Mendonça IC, Porto IC, do Nascimento TG, de Souza NS, Oliveira JM, Arruda RE, Mousinho KC, dos Santos AF, Basílio-Júnior ID, Parolia A, Barreto FS. Brazilian red propolis: phytochemical screening, antioxidant activity and effect against cancer cells. BMC Complementary and Alternative Medicine. 2015 Dec;15:1-2.

Answer: The requested articles have been duly incorporated into the introduction. The studies by Rufatto et al. (2018) and de Mendonça et al. (2015) are cited in Introduction section, page 3, line 50. Additionally, the references to Silva et al. (2022) and Aldana-Mejia et al. (2021) are included in Introduction section, page 3, line 56.

7- In line 88, add missed solvent (hexane: 70:30) and

Answer: The solvent was added (page 5, line 92).

8- The font size and style in lines 103 to 106 should be the aligned with all the text.

Answer: The formatting has been adjusted as requested.

9- Why tetramethylbenzene (TMB) was used as internal standard? mention the reasons in section 2.4.2

Answer: The TMB was chosen as the internal standard because it is chemically stable and does not interfere with the detection of the analytes, as its retention time is sufficiently distinct. Additionally, TMB is commercially available in high purity, making it a reliable and reproducible standard. These characteristics make TMB an appropriate choice for internal standardization in our method. This information was added to the manuscript (page 7, lines 133-135).

10- In line 181, the date for ethical approval should reported.

Answer: The date of ethical approval has been added as requested (page 9, lines 187-188).

11- Write full names for all abbreviations when first mentioning e.g. EDTA, DMH

Answer: The full names for the abbreviations had been included in the text. Specifically, the full name for DMH can be found in page 12, line 251, and for EDTA in page 13, line 253.

12-In depth discussions with comparisons with other types of propolis are strongly recommended?

Answer: Propolis composition is highly variable, depending on several factors, such as the geographic location of the hives, season of the year, and characteristics of the local vegetation (Šturm and Ulrih, 2020). Propolis has no “typical” composition; on the contrary, it is well known to vary widely from one type to another and from region to region (Wieczorek et al., 2022; Salatino, 2022).

Considering Brazil’s continental dimension and its conspicuous plant diversity, several Brazilian types of propolis have been classified, including color, botanical source, and phytochemical profile (Park et al. 2002). The different types of propolis bear different chemical profiles, leading to different chemical compositions and, consequently, different biological activities. For example, an inverse proportion of the contents of triterpenoids versus phenolic substances is observed in green propolis derived from Baccharis dracunculifolia in Bolivia and Brazil (Nina et al., 2016; Salatino et al., 2021). Propolis containing high phenolic contents are likely to have high antioxidant and antibacterial activity, in contrast with propolis of the same type with high contents of triterpenoids and beeswax and low amounts of phenolic components.

In view of the above, the results obtained in the present study were not compared with other types of propolis.

Nina, N.; Quispe, C.; Jimenez-Aspee, F.; Theoduloz, C.; Gimenea, A.; Schmeda-Hirschmann, G. Chemical profiling and antioxidant activity of Bolivian propolis. J. Sci. Food Agric. 2016, 96, 2142–2153.

Park, Y.K; Alencar, S.M.; Aguiar, C.L. Botanical origin and chemical composition of Brazilian propolis. J Agric Food Chem. 2002, 50, 2502–2506.

Salatino, A. Perspectives for uses of propolis in therapy against infectious diseases. Molecules 2022, 27, 4594.

Salatino, A.; Salatino, M.L.F.; Negri, G. How diverse is the chemistry and plant origin of Brazilian propolis? Apidologie 2021, 52, 1075–1097

Šturm, L.; Ulrih, N.P. Advances in the propolis chemical composition between 2013 and 2018: A review. eFood 2020, 1, 24–37.

Wieczorek, P.P.; Hudz, N.; Yezerska, O.; Horˇcinová-Sedláˇcková, V.; Shanaida, M.; Korytniuk, O.; Jasicka-Misiak, I. Chemical variability and pharmacological potential of propolis as a source for the development of new pharmaceutical products. Molecules 2022, 27, 1600

13-Abbreviation list is strongly recommended

Answer: A list of abbreviations was not included, since the author guidelines provided by the journal do not specify this requirement. However, should the journal permit or recommend the inclusion of such a list, we would be pleased to provide it.

14- Add future plans and study limitation as one paragraph at the end of the discussion or the conclusion

Answer: Future plans and study limitations have been inserted, as requested (page 33, lines 639 and 640; page 34, lines 641-643).

15- Conclusion should be written with full names for all abbreviations HPLC-PDA, BFRP, and RPHE.

Answer: The conclusion has been revised to include the full names for all abbreviations, as requested (page 33).

Reviewer 2 Report

Comments and Suggestions for Authors

The paper entitled: "Development of a New Red Propolis Extract and Evaluations of its Toxicity and Efficacy Against Colon Carcinogenesis" is a well-written draft that investigatesthe potential therapeutic benefits of red propolis, particularly focusing on its application in colon cancer treatment. In my opinion, it is suitable to be published in “Pharmaceuticals” after considering following points.

1. The abstract should be constructed in a graphical figure depicting extraction and development of red propolis, then testing and evaluation and key outcomes.

2. The manuscript could elaborate more on the specific mechanisms of action by which red propolis exerts its anticancer effects.

3. Compare between red propolis and existing cancer treatments, so that the proper advancement can be understood by the readers.

Also mention how the natural origin of red propolis may appeal to patients seeking alternative treatments with fewer side effects.

Overall, this paper presents a significant contribution to the field of natural product-based cancer therapies. The development of a new red propolis extract and its promising results against colon cancer highlight the potential of propolis as a safe and effective treatment.

What is the main question addressed by the research?
The study addressed the main active constituents of red propolis extract along with its effects against specific types of cancer. Along with this the extract used is properly standardized.

Do you consider the topic original or relevant to the field? Does it address a specific gap in the field? Please also explain why this is/ is not the case.
Yes, though some work is already done related to the current manuscript but the current manuscript is focussed on colon cancer.

What does it add to the subject area compared with other published papers material?
Most of the studies reported are not focussed one, albeit the current study is specific and in an organized manner.

Are the conclusions consistent with the evidence and arguments presented and do they address the main question posed? Please also explain why this is/is not the case.
Yes, They performed sufficient studies to support their hypothesis.

Are the references appropriate? Yes

Any additional comments on the tables and figures":
The abstract should be constructed in a graphical figure depicting extraction and development of red propolis, then testing and evaluation and key outcomes.

Comments on the Quality of English Language

NA

Author Response

We would like to thank the Reviewer for all the insightful and constructive comments on our work. Each comment is addressed below in a point-by-point way, as well as all changes made on the manuscript are highlighted in yellow for better understanding.

The paper entitled: "Development of a New Red Propolis Extract and Evaluations of its Toxicity and Efficacy Against Colon Carcinogenesis" is a well-written draft that investigates the potential therapeutic benefits of red propolis, particularly focusing on its application in colon cancer treatment. In my opinion, it is suitable to be published in “Pharmaceuticals” after considering following points.

1- The abstract should be constructed in a graphical figure depicting extraction and development of red propolis, then testing and evaluation and key outcomes.

Answer: Thanks for your suggestion. A graphic summary was constructed to represent the development process of the benzophenones-free red propolis extract, followed by testing, evaluation and presentation of the main results.

2- The manuscript could elaborate more on the specific mechanisms of action by which red propolis exerts its anticancer effects.

Answer: Thank you for the insightful suggestion. We acknowledge the importance of elucidating the specific mechanisms by which red propolis exerts its anticancer effects. However, the primary objective of this study was focused on the development of a novel red propolis extract with reduced toxicity and significant chemopreventive potential. Our intention was to evaluate its safety profile and potential efficacy as a therapeutic agent, rather than to delve into the molecular mechanisms underlying its anticancer activity. We appreciate your recommendation, and future studies will aim to investigate these mechanisms in greater detail to complement our findings.

3- Compare between red propolis and existing cancer treatments, so that the proper advancement can be understood by the readers. Also mention how the natural origin of red propolis may appeal to patients seeking alternative treatments with fewer side effects.

Answer: Thank you for your feedback. We have addressed your comment by including a comparison between red propolis and existing cancer treatments in the manuscript. This section highlights how red propolis offers a less toxic alternative, compared to the systemic toxicity often seen with conventional therapies (page 3, lines 59-63; page 4, lines 64-73).

Reviewer 3 Report

Comments and Suggestions for Authors

Article is curious and important. However, it should be supplemented by additional data and contain some editing errors. My issues were listed below:

Major issues

- Why authors did not compare benzophenones-free red propolis extract with hydroalcoholic extract of red propolis by anyone liquid chromatography techniques such as LC-MS or LC-DAD? In the figure 2A, chromatogram had a lot of minor peaks. Authors should determine if they were not benzophenones. At the present stage is not possible to claim, that authors obtained benzophenones-free red propolis extract. Authors should perform additional HPLC-DAD or HPLC-MS analysis and mark benzophenones peaks.

- Authors claimed, that they obtained “45.4% yield of BFRP”. Extraction yield was calculated for the crude propolis or residues after hexane extraction?

- Please explain in discussion section, what is “long term using of propolis” and when it may be potentially dangerous for humans. Is this include external, internal using of propolis or both?

Minor issues:

- Some figures were illegible (ex. figure 3, 4 and 5). Please correct them.

- Tables should fit on one page (if possible). Please correct them.

Author Response

We would like to thank the Reviewer for all the insightful and constructive comments on our work. Each comment is addressed below in a point-by-point way, as well as all changes made on the manuscript are highlighted in yellow for better understanding.

Article is curious and important. However, it should be supplemented by additional data and contain some editing errors. My issues were listed below:

Major issues

  1. Why authors did not compare benzophenones-free red propolis extract with hydroalcoholic extract of red propolis by anyone liquid chromatography techniques such as LC-MS or LC-DAD? In the figure 2A, chromatogram had a lot of minor peaks. Authors should determine if they were not benzophenones. At the present stage is not possible to claim, that authors obtained benzophenones-free red propolis extract. Authors should perform additional HPLC-DAD or HPLC-MS analysis and mark benzophenones peaks.

Answer: The hydroalcoholic extract of red propolis has been extensively studied in previous works by our research group, providing us with a comprehensive understanding of its chemical composition. The minor peaks observed in the chromatogram of the benzophenones-free red propolis extract correspond to other known phenolic compounds found in red propolis, such as calycosin, formononetin, and biochanin A, among others. These phenolic compounds were not included in the method validation due to the limited quantities isolated, which were insufficient for the entire validation process. Given the physicochemical characteristics of benzophenones and the C18 column used, a long-term isocratic elution with 100% organic mobile phase would be necessary to detect benzophenones. Mejia et al. (2021) developed a 50-minute method where a mixture of benzophenones (guttiferone E/xanthochymol and oblongifolin A) eluted after 46 minutes (Doi: 10.1016/j.jpba.2021.114029). In another study, a 105-minute method was required to separate benzophenones, with guttiferone E and xanthochymol eluting at 79 minutes and oblongifolin B at 80 minutes (Doi: 10.1021/acs.chemrestox.0c00356). The absence of benzophenones allowed us to develop a more time-efficient method, completing the analysis in 30 minutes. We include an analysis of the hydroalcoholic extract and the benzophenones-free red propolis extract by HPLC-DAD in the supplementary material.

2- Authors claimed, that they obtained “45.4% yield of BFRP”. Extraction yield was calculated for the crude propolis or residues after hexane extraction?

Answer: The extraction yield of 45.4% for BFRP was calculated based on the crude red propolis weight prior to hexane extraction.

3- Please explain in discussion section, what is “long term using of propolis” and when it may be potentially dangerous for humans. Is this include external, internal using of propolis or both?

Answer: In response to the reviewer's comment, the discussion section has been revised (page 29, 550-558).

Minor issues:

4- Some figures were illegible (ex. figure 3, 4 and 5). Please correct them.

Answer: Necessary corrections have been made to Figures 3, 4, and 5 to improve their legibility.

5- Tables should fit on one page (if possible). Please correct them.

Answer: Tables have been adjusted to fit on one page where possible.
